# Jet Grouping of Linear-Shaped Charges and Penetration·Performance

**Seokbin Lim** [1,*] **, Philipp Baldovi** [2] **and Christopher Rood** [1]

1 Energetic Systems Research Group, Department of Mechanical Engineering, New Mexico Tech, Socorro, NM 87801, USA

2 Test and Evaluation Division, Detonation and Combustion Technology Branch, Naval Surface Warfare Center, Indian Head, MD 20640, USA

* Correspondence: bin.lim@nmt.edu

**Abstract:** There have been many studies on the penetration performance of conical-shaped charges (CSCs) in various applications. These studies have led to a great deal of improvement, and even breakthroughs, in the design of CSCs. These studies have also positively affected the design of linear-shaped charges (LSCs), but due to the comparatively low penetration performance of LSCs there has been little theory-based scientific study of their penetration performance. In this paper, empirically observed field testing data are presented, including the details of the penetration performance and configuration of the LSC jet against UHMW-PE (ultra-high molecular weight polyethylene) targets using a series of flash X-ray exposures. These findings are compared to the previously published theoretical analysis in order to understand the nature of the formation, penetration, and classification/grouping of LSC jets. After a series of empirical and theoretical investigations, it was concluded that the LSC jet can be grouped depending on the jet particulation and distribution pattern, and each jet group exhibits different penetration performance. The relationship between liner collapse angle and jet segment density suggests potential room for performance improvement.

**Keywords:** linear-shaped charges; jet grouping; penetration performance; X-ray images

## 1. Introduction

Scientific studies of LSCs have been conducted in various research communities for many years, including the military, space, and commercial sectors. The main research interest has been the penetration performance of LSCs, since, except for specially designed large LSCs, they exhibit poor cutting performance of a matter of several inches (approximately 20~50 mm) at maximum. This tendency to low penetration becomes more apparent when the penetration depth is compared with CSCs, where the penetration performance is significantly better. There have been many engineering attempts to overcome the limited penetration performance of LSCs using various approaches including novel liner materials or liner configurations [1–3]. Unfortunately, while there have been considerable advances, there have been very few meaningful breakthroughs in this field of study, because the jet formation process of LSCs has not been fully studied or understood. This is due to the complex nature of the jet formation during the collapse of the LSC liner. Note that the fundamental LSC jet formation process is different from that of CSCs due to the different geometry and detonation front propagation [4].

There have been multiple attempts to observe the nature of jet projection of LSCs using the flash X-ray imaging technique. The initial attempts were made by P. Cooper and J. Stofleth (unpublished), and these images elicited a great deal of interest in the scientific community because it was the first time the jet projection pattern of LSCs had been imaged (commercially manufactured 600 and 900 g/ft LSCs) [5,6]. Note that the unit of g/ft has been used in this field of study as customary, and 1 g/ft = 212.6 mg/m.

These images have revealed several unique features typical of LSCs. First, no solid jet was observed before particulation. The liner collapse of the LSC produces severely particulated jet segments from the start without any solid jet. Second, the particulated LSC jet segments always project in a line-grouping pattern, and each jet group has its own angle of projection, due to the axial directional detonation propagation in the LSC charge [5–7].

In order to explain the nature of LSC jet projection, Lim proposed several engineering approaches, along with supplemental numerical simulation. Lim identified several unique features of LSCs in numerical simulation, including the LCL (liner collapse line) [5]. The LCL is a collection of liner collapse points parallel to the detonation propagation direction. Note that in CSCs there is only a single liner collapse point because detonation is axially symmetrical. The LCL provides a clue to the LSC jet grouping trends [5,6].

Lim also identified the flight pattern of the LSC jet. Because each LSC jet group has different jet projection properties, including different angles, the jet does not tend to project at a right angle. Instead, an LSC jet projects slightly forward, making it hard to aim if the flight to target is long. This hinders proper design of special LSCs where a long flight before the hit-on-target is required; the problem has been investigated experimentally (unpublished) [5].

A three-dimensional configuration and analysis approach has been adopted for theoretical study of the jet formation process of LSCs, although a two-dimensional approach would be reasonable for the theoretical analysis of CSCs [8,9]. This is because the jet formation and projection direction of the LSC is not in-line (or not parallel) with the detonation direction.

In terms of the theoretical study LSCs, most research in LSCs has been inspired by the work of Birkhoff, who first addressed the theory of shaped charges [10]. Birkhoff's theory assumes a steady-state condition; however, a more advanced Pugh–Eichelberger–Rostoker (PER) theory was developed later [11]. The PER theory allows more realistic and explicit options for the study of the jetting behavior of general-shaped charges; it treats the non-steady state collapse of the liner and is more applicable to the real shaped-charge design. While the Birkhoff and PER theories are focused on CSCs, there have been multiple attempts to apply them to LSCs.

Lim proposed a rather simple steady-state LSC theory based on the Taylor bending angle [4]. This approach offers a way to predict the functioning of LSCs based on a three-dimensional approach while maintaining a linear ideal collapse assumption. Lim also proposed another steady-state LSC theory using a simple trigonometry and it delivers identical results; however, it was retracted later due to the issue in the simulation and potential duplication of the analysis. These LSC jet formation theories are limited by the steady-state assumption. A non-linear (or non-steady state) jet formation theory based on an arc deformation of the flat liner has also been proposed [12]. These research efforts, combined with the newly proposed LCL, allow a more advanced understanding of LSCs.

In summary, there have been publications and research efforts in LSCs, but most results still have not brought meaningful breakthrough in terms of jet penetration performance. In addition, there have been no notable research efforts on the combination of the jet formation theory and the LSC the cutting process, mainly due to the technical difficulty of observing the jet formation and cutting process in (or near) a target. This paper aims therefore to uncover some of the jet formation and cutting processes of LSCs based on a series of field tests and theoretical calculations.

## 2. Objectives

As noted above, the theoretical jet formation and cutting processes of LSCs have rarely been studied together due to the technical difficulty of the problem. The LSC jet in the middle of the cutting process (or the jet-on-target) is not easy to observe, and it is difficult to connect the jet formation theory with the cutting performance. In particular, the equation of motion of the jet will not provide a detailed prediction regarding the penetration since the equation of motion of the jet only predicts a few parameters that affect the penetration performance, and many more must be identified for proper prediction.

A typical shaped-charge jet (CSCs and LSCs) produces a finite length of jet. Due to significant jet elongation during flight (with a different jet velocity profile along the length of the jet), each individual jet segment will have its own unique penetration capability. This fact is more significant for LSCs because there is no solid jet stage before the particulation.

The main objective of the present research is to understand the penetration capability of each segment (or group) of a particulated jet, and this will be analytically studied in a manner that traces back to the original liner collapse point condition. This information will help to illuminate the penetration capability of LSCs.

It is important to note that the penetration capability of each segment must be identified in the middle of penetration. Otherwise, all the jet segments hit the target and there is no way to find the penetration capability per segment.

### 3. LSC Test and Observation

As an initial attempt to see the relationship between jet formation theory and the cutting performance per each segment, a series of field tests using flash X-ray was carried out.

Multiple 18-in.- (or 457.2-mm-) long 2000 g/ft. commercially manufactured copper-lined LSCs (with a flat liner) were shot against an X-ray-transparent material of UHMW (ultra-high molecular weight polyethylene), density $\rho = 0.95$ g/cm$^3$, $24 \times 12 \times 5$ in. (or $609 \times 305 \times 127$ mm) block, and the penetration event was captured using a single 150 kV flash X-ray system located 4 feet (1219 mm) away (Figure 1). The geometrical specifications and the core charge properties of the 2000 g/ft LSCs are listed in Table 1 [5]. Detailed dimensional specification of the LSC charge is listed in [5].

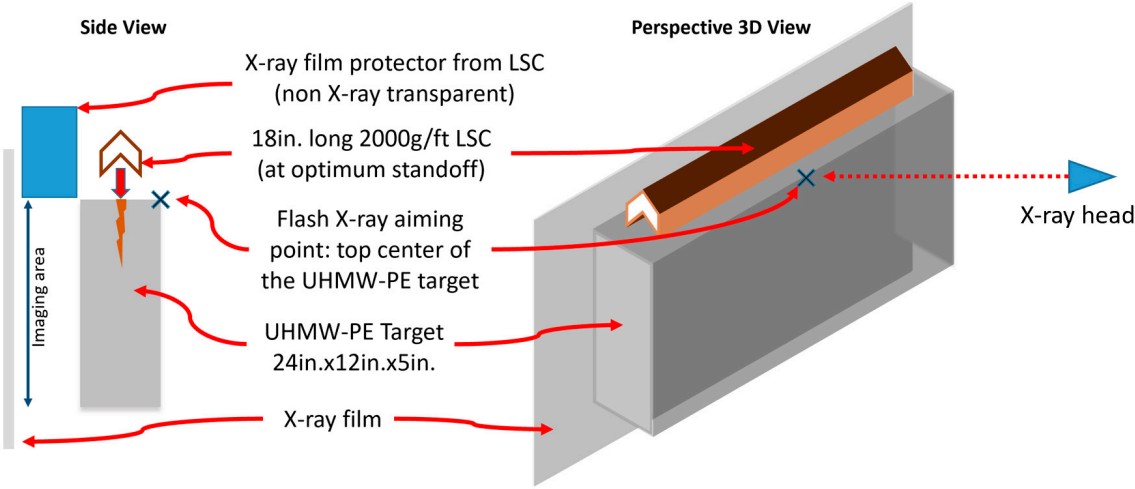

**Figure 1.** Schematic diagram of the LSC test configuration.

Because UHMW is transparent under the flash X-ray and the copper jets are not, the use of UHMW as a target material will reveal the penetration history during the jet-on-target. In order to improve the image quality and to prevent technical difficulties, multiple LSC shots with different exposure times were used to reveal the penetration time history in the target. It was assumed that the same sized LSCs would provide identical penetration or jet projection patterns. Previous field tests have shown that LSC detonation has quite reasonable consistency between shots (as long as the size of the LSC is identical), and this approach was determined to be reasonable.

To capture the penetration time history in the UHMW target, the timing of the X-ray exposure is critical. Based on the calculated jet speed of the LSCs and an estimated penetration speed in the target, three exposure times of 75, 105, and 165 μs after detonation were selected and three X-ray images were produced (Figure 2).

**Table 1.** Properties of LSCs and the mechanical properties of UHMW targets [5].

| | Category | Values | Unit |
|---|---|---|---|
| Liner (Copper) | Density | 8.90 | g/cm$^3$ |
| | Thickness | 1.65 | mm |
| | Height ($l$) | 18 | mm |
| | Apex angle ($\alpha$) | 36 | degree |
| Explosives (HMX) | Density | 1.6 | g/cm$^3$ |
| | Thickness | 6.1 * | mm |
| | Height | 18 | mm |
| | CJ detonation vel. | 9.11 | mm/µs |
| Target ** (UHMW) | Density | 0.95 | g/cm$^3$ |
| | Tensile Strength | 39~48 | MPa |
| | Yield Strength | 21~28 | MPa |

* An average value. ** Static material properties. Dynamic material properties are not available.

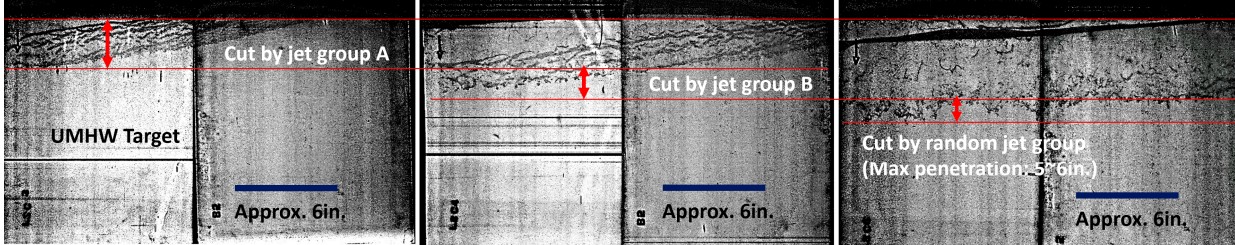

**Figure 2.** Flash X-ray images of 2000 g/ft LSCs penetrating UHMW targets. The LSC is located above the frame, and is not shown. From left to right, 75, 105, and 165 µs after initiation. The detonation front moves from left to right in each image. White area is UHMW, and particulated LSC jets (black-colored segments in upper middle) are moving downward, penetrating the UHMW target.

The three X-ray images show interesting patterns in jet during penetration. As in the X-ray images of Stofleth and Cooper, the LSC jet showed a well-organized pattern of particulation, depending on time after detonation (Figure 3). Based on the distribution density and formation of jet particles in the X-ray images, it was able to separate the jet segment groups into three categories: high-density regular jet (Jet Group A), low-density irregular jet (Jet Group B), and random jet particles (behind B, where large slugs are included). A detailed description follows (Figure 3).

In Jet Group A (high-density jet) all jet segments were close together and of uniform size/pattern distribution. Jet Group B showed a somewhat organized pattern, but the distance between the jet particles was larger than in Jet Group A and there were fewer jet particles. The size of each jet particle in B was much bigger than those in Jet Group A. Random jet particles followed behind Jet Group B, but there was no clear evidence that this group was well-organized, and the number of jet segments was not significant (bottom image in Figure 3). The penetration progressed in sequence: Jet Group A, Jet Group B, and then the random particles.

Close observation of the images indicates that the interface between the jet (black dots and lines) and UHMW target (grey area) shows two distinct slope angles, depending on the speed of penetration, slow or fast (red-dotted slope lines in Figure 3). The fast cutting occurred in the very early stage of penetration (mainly Jet Group A), and the slow cutting occurred in the later stage (tail of Jet Group A, but principally Jet Group B, top and middle figures in Figure 3). Note that the slope lines indicate the speed of penetration, since the detonation front moved from left to right. In other words, the time history of the penetration is shown in a single X-ray image, from left to right. The disappearance of jet segments indicates the consumption of jet segments during penetration. By pinpointing the location of the jet segments' disappearance, the penetration depth of each jet group can be identified. Near the end of effective penetration by Jet Groups A and B, a large slug is

moving into the target (a large black line in the upper part of the bottom figure in Figure 3), but the large slug does not contribute to the effective penetration and is not addressed here.

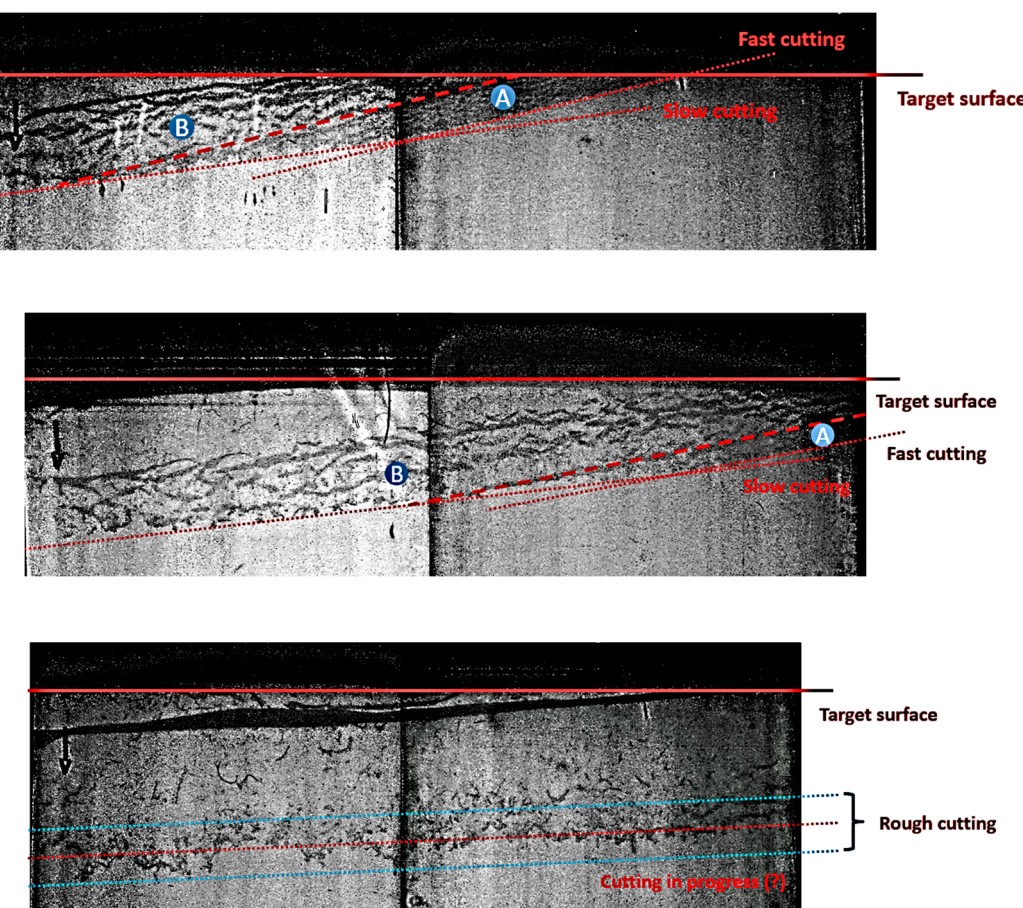

**Figure 3.** Magnified flash X-ray images of LSC jet penetration into UHMW blocks showing 75 μs (**Top**), 105 μs (**Middle**), 165 μs (**Bottom**) after initiation (detonation proceeds left to right). Red dotted lines represent cutting/penetration line and grouping of jet particles.

Maximum penetration depth in the UHMW targets was measured around 5~6 in. (127~153mm). Note that the exact measurement of the maximum penetration was not possible, as the bottom of the penetrated section is not cleanly cut.

From this information and direct measurement from the series of X-ray images, it was concluded that the penetration depth driven by Jet Group A was around 40~50%, and Jet Group B contributed around 30~40% of the total penetration. While Jet Group A shows the most favorable penetration, the random final jet group shows only around 10% of the total penetration.

In conclusion, multiple jet groups were observed in the X-ray images, including: high-density regular jet (Jet Group A), low-density irregular jet (Jet Group B), and random jet particles. Jet Group A showed the most favorable penetration depth and speed. Because each group formed in a different time step, the jet grouping was dependent on the liner collapse process, and each liner collapse stage would produce a different jet group (different properties) resulting a different penetration performance.

Note: the estimated penetration depth was calculated based on the assumption that the state at 165 μs after detonation revealed the maximum penetration depth in the given target.

## 4. Theoretical Analysis

Lim previously proposed a theoretical study of the LSC liner collapse based on the arc deformation of the LSC liner [12]. This theory is based on the assumption that the flat

LSC liner will deform in an arc during projection, meaning that the collapse-point angle cannot be identical to the original apex angle. In order to determine the non-linear liner collapse-point angle $\beta_t$, the following schematic diagram was used (Figure 4).

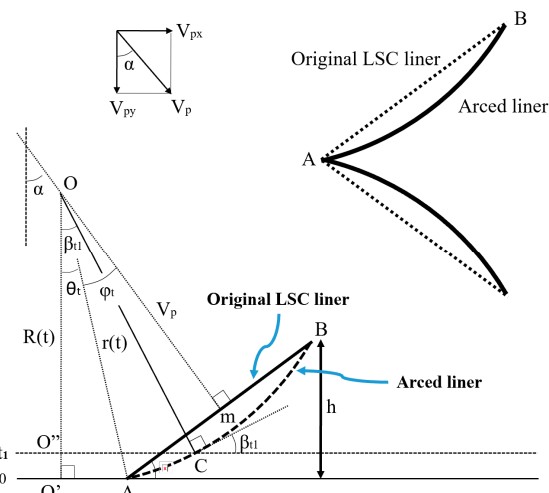

**Figure 4.** Schematic diagram of arc-deformed LSC liner collapse [12].

Where $\alpha$ is the original liner apex half-angle, $\beta_t$ is a liner collapse-point angle, $V_p$ is the liner velocity, $V_{py}$ is the vertical component of the liner velocity, $h$ is half of the LSC base width, $r(t)$ is the radius of the flyer liner arc, and $R(t)$ is the vertical height of the arc center.

Based on the diagram above, the following collapse-point angle equation is derived [12].

$$\beta_t = cos^{-1}\left(\frac{R(t) - V_{py}t}{r(t)}\right)$$

In order to achieve the arc deformation of the flat liner, Lim used a hydrocode simulation to realize the arc deformation. Later, the flat liner arc deformation was experimentally measured, and it showed favorable agreement with the hydrocode simulation in unpublished data [13,14].

According to the arc-deformed liner collapse theory, upon detonation, the two sides of the flat liner start moving toward each other to collapse. Due to the strong release of detonation gas along the edges of the liner, the edge of the liner moves slower than the center of the liner, and this causes an arc deformation [12–14]. This gradual arc deformation, followed by the collapse of the two arc-deformed liner sides, the collapse- point angle varies in a unique trend (Figure 5).

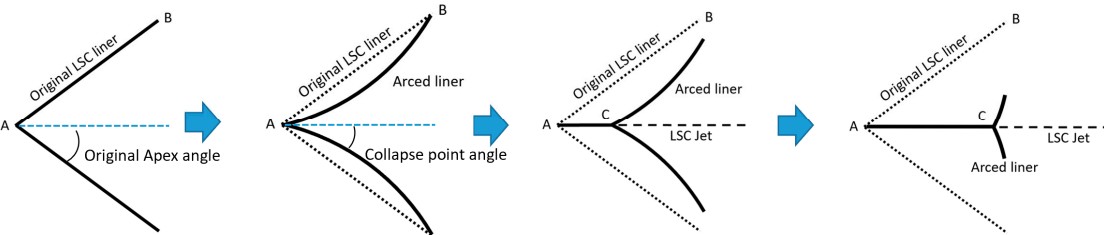

**Figure 5.** LSC liner collapse progress with arc deformation [13].

As is schematically illustrated in Figure 5 above, when the two arc-deformed liners approached each other in collapse, the collapse-point angle initially decreased (second from left, Figure 5), but as the collapse progressed the collapse-point angle gradually increased. Ultimately this angle approached a point where the formation of the jet was no longer effective (right, Figure 5).

The collapse-point angle can thus be divided into two areas, depending on whether it is larger or smaller than the original apex angle, or favorable or non-favorable for jet

formation. This is an important characteristic of the LSC jet; the collapse point angle is an important parameter that controls the jet property [4]. The gradual change of the collapse- point angle provides a clue to the jet grouping described in the previous section. Jet speed variation will control the jet stretching, affecting particulation (high-/low-density jet), and eventually provides a clue to the difference in jet segment density in Jet Groups A and B. In order to illuminate the relationship of jet velocity variation and collapse-point angle change, the arced liner radius and calculated parameters were tabulated, and the following equations from the original Birkhoff theory were used to calculate the jet velocity (Figure 4), [12].

$$V_1 = \frac{V_p}{sin\beta_t} sin\left\{90 - \left(\frac{\beta_t - \alpha}{2}\right)\right\}, \text{ and } V_2 = V_p \left\{\frac{cos\left(\frac{\beta_t - \alpha}{2}\right)}{tan\beta_t} + sin\left(\frac{\beta_t - \alpha}{2}\right)\right\}$$

where $V_j = V_1 + V_2$, and $V_s = V_1 - V_2$.

The following data were collected from Lim [12] (Table 2, Figures 6 and 7).

**Table 2.** The 2000 g/ft arced liner radius $r(t)$ and calculated parameters [12].

| Time (μs) * | 0 | 1 | 2 | 3 | 4 | 5 | 6 | 7 | 8 | 9 | 10 | 11 | 12 |
|---|---|---|---|---|---|---|---|---|---|---|---|---|---|
| $r(t)$ (mm) | ∞ ** | ∞ ** | 61.66 | 47.09 | 32.52 | 24.93 | 22.04 | 18.99 | 16.59 | 15.81 | 14.12 | | |
| $\beta$ (°) | | | 31.47 | 32.73 | 35.50 | 40.42 | 45.59 | 52.16 | 59.69 | 65.37 | 74.17 | N/A *** | |
| $V_j$ (mm/μs) | | N/A | 4.439 | 4.272 | 3.949 | 3.484 | 3.107 | 2.738 | 2.419 | 2.229 | 1.996 | | |
| $V_s$ (mm/μs) | | | 0.406 | 0.408 | 0.411 | 0.417 | 0.424 | 0.435 | 0.451 | 0.465 | 0.490 | | |

* Time after detonation. ** Too large to measure. *** LSC liner collapse completed, violating the physically possible collapse range ($h \geq V_{py}t$). Note: Original apex angle was 36°, $r(t)$: radius of flyer liner arc, $\beta$: liner collapse point angle, $V_j$: jet velocity, $V_s$: slug velocity.

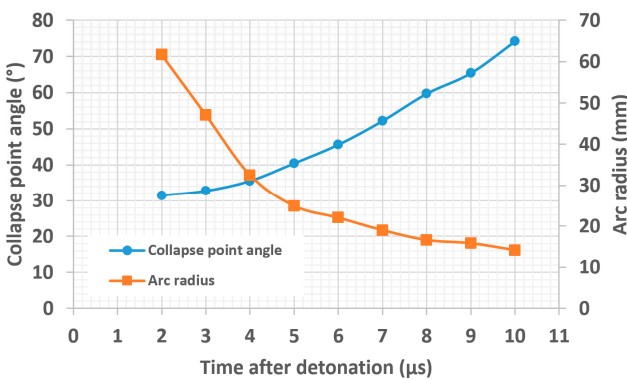

**Figure 6.** Collapse-point angle and arc radius after detonation [12].

Figure 6 shows a sudden change of the liner arc radius at around 4~5 μs after detonation. The arc radius decreased at a higher rate until 4~5 μs after detonation, mainly driven by the strong detonation energy right next to the flat liner. This high rate of arc deformation in the early stage of detonation has been studied previously [13], and a very similar trend was seen. After this initial high rate of arc deformation, the arc deformation slowed, as the detonation gas behind the liner lost expansion energy. The arc deformation, the change of radius, after 5~6 μs became very slow. This clearly indicates that the effect of the detonation gas behind the liner weakened as the liner moved away.

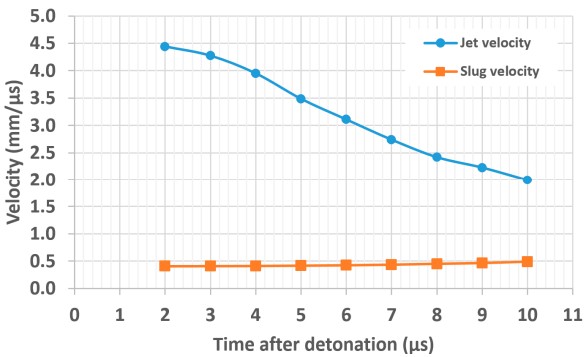

**Figure 7.** Jet and slug velocity of 2000 g/ft LSC [12].

The distinct transition of arc deformation rate (from fast to slow deformation rate) near 4~5 µs is not yet clearly understood. It relates to the transition from the hydrodynamic regime to the typical plastic/elastic deformation of the liner, while the detonation gas pressure decreases very fast during expansion. This point requires more detailed investigation, and is out of the scope of this paper.

The deformation behavior described here, with distinct transition from angles smaller than to larger than the original collapse point angle, provides a clue to why the LSCs formed unique Jet Groups A and B. In Figure 6, for example, the original LSC apex angle was 36°. Before 4~5 µs after detonation, the collapse-point angle was below 40° and the rate of increase of the collapse point angle was low (slope Figure 6). This provided a favorable jet formation condition and the jet velocity reached maximum speed. After 4~5 µs, the collapse-point angle increased rapidly and eventually reached a large angle (non-favorable jet formation condition). Figure 7 shows that the jet velocity decreased rapidly after the same point.

That is, there was a clear difference in jet formation pattern before and after 4~5 µs after detonation. Jet Groups A or B are not precisely visible in the theoretical analysis above, because the liner collapse was not observable in the X-ray images, but there was a clear transition point during the jet formation where the jet properties changed. In addition, if one considers the jet speed difference per time step before and after 4~5 µs, there was a clear difference, producing different jet behaviors.

The jet velocity change trend during liner collapse (or jet projection) will provide a clue to understanding jet particulation. If the jet velocity difference per time ($dV/dt$) is not large (Figure 8), then the jet segments tend to stay together, remaining densely spaced.

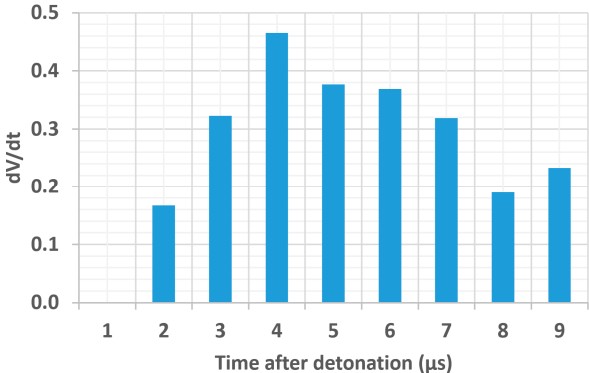

**Figure 8.** Jet velocity difference per time.

According to the $dV/dt$ values derived from Table 2, the $dV/dt$ was comparatively low in the early stage. Before 4 µs, the average was 0.245. After this point, the $dV/dt$ grew to an average $dV/dt$ of 0.325, reflecting the distinct transition before and after 4 µs.

In other words, the liner collapse condition before 4 µs created a condition in which the jet segments stayed close each other (Jet Group A). After 4 µs, the jet speed difference per time step was increasing, making the jet segments spread apart significantly after 4~5 µs (Jet Group B).

## 5. Discussion

X-ray transparent UHMW targets were used here to identify the penetration history of LSCs. X-ray transparent materials are by nature low in density, and this will affect penetration performance and may suggest questions about the present analysis. High-strain- rate dynamic conditions are typically non-linear, so the penetration performance for each jet group might be different against other types of target materials, and a different type of target material might produce slightly different results.

The theoretical analysis addressed above is based on the location where the jet is created. In other words, all the data in Table 2 are based on the liner collapse point, and this is invisible in the X-ray experimentation. In addition to this, the jet created from the liner collapse point needs to fly around several inches (up to 6 inches, or 152mm, into the target) before the flight pattern is captured by the X-ray film. This long-distance jet flight will create a large jet speed discrepancy between the theoretical calculation and the experimental measurement due to the severe jet speed reduction during flight. In order to handle this discrepancy, the comparison between the experimentation and the theoretical analysis was accomplished based on the following assumptions: (1) the LSC jet tip in the experimentation (or X-ray images) originates from the 2 µs after detonation in Table 2; (2) the LSC jet slug in the experimentation (or X-ray images) originates after 10 µs after detonation in Table 2. This assumption can be made because the data in Table 2 is designed to indicate the start and end point of the liner collapse.

Based on these assumptions, the three X-ray images are connected and the corresponding theoretical data from the Table 2 are compared (Figure 9).

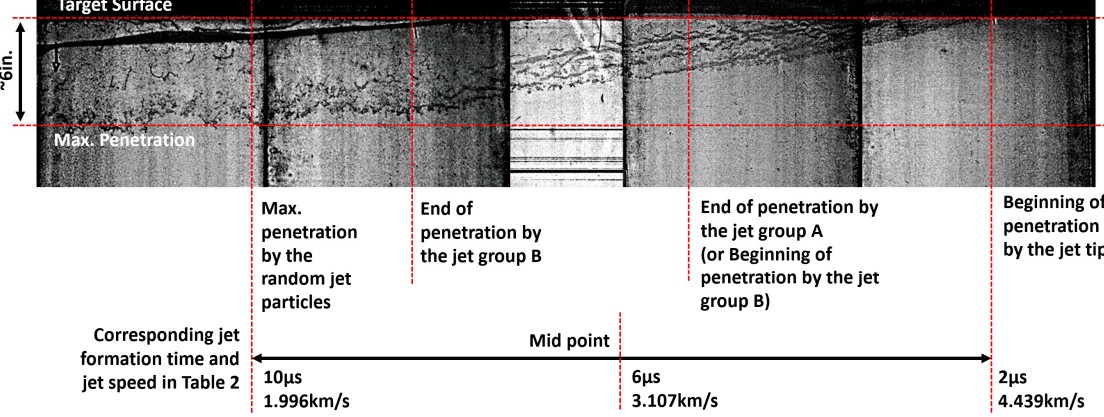

**Figure 9.** Connected X-ray images and corresponding jet properties at the liner collapse point.

In the present paper, the separation of Jet Group A and B was derived from the theoretical analysis based on the jet projection speed and the liner arc deformation rate. Both of these parameters are closely related to many different dynamic parameters during the jet formation and the extreme dynamic process of liner collapse. One of the interesting observations from the X-ray images is the distinct separation between Jet Groups A and B. This clear distinction indicates a significant and drastic change of the dynamic properties of the liner right at that collapse moment. If the clear distinction between Jet Groups A and B is related to the extreme dynamic properties of the liner materials, then the use of different liner materials would deliver an interesting result and should be investigated further.

## 6. Conclusions

A series of X-ray images shows different jet formation groups, depending on jet particle distribution pattern and density. This includes a high-density regular jet (Jet Group A), a low-density irregular jet (Jet Group B), and random jet particles. Because these groups form at different times and at different stages of liner collapse, one can conclude that the liner collapse condition controls the jet grouping.

Each jet group penetrates a given UHMW target at a different speed and depth. That is, the high-density regular jet shows better penetration performance in terms of depth and speed and accounts for approximately 40~50% of effective penetration. The low-density irregular jet shows efficacy, with depth of penetration of around 30~40% of the entire penetration after 165 μs, although the speed of penetration is slower than that of the high-density regular jet. Random jet particles and a large slug exhibit minimal penetration, and it is excluded from this study.

This series of observations regarding penetration performance is compared to a theoretical analysis, which indicates that there is a unique collapse condition change near 4~5 μs after detonation. This is driven by the liner deformation rate and the detonation gas pressure. The liner deformation rate depends on the detonation gas expansion behind the liner, followed by rarefaction intrusion into the center of the liner. The detonation gas expansion followed by liner deformation affects the liner collapse-point angle, and this will be the main driver varying the jet particulation patterns and jet grouping.

Due to the technical limitation of the data collection during the X-ray experimentation, the direct comparison between the theoretical calculation and the experimental observation was not accomplished. However, based on simple assumptions regarding the origination of the jet driven by the liner collapse and the jet penetration starting/ending point in the target, the penetration performance behavior per each jet segment was estimated.

**Author Contributions:** Conceptualization, S.L.; methodology, S.L.; validation, S.L. and P.B.; theoretical analysis, S.L., P.B.; Experimentation, S.L. and P.B.; data analysis, C.R. and P.B.; writing—original draft preparation, S.L.; writing—review and editing, C.R. and P.B.; supervision, S.L.; project administration, S.L.; funding acquisition, S.L. All authors have read and agreed to the published version of the manuscript.

**Funding:** This research was funded by Office of Naval Research grant number N00014-12-1-0377, and The APC was funded by New Mexico Tech.

**Informed Consent Statement:** Not applicable.

**Conflicts of Interest:** The authors declare no conflict of interest.

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
