# Peer review of "Jet Grouping of Linear-Shaped Charges and Penetration Performance"

_applsci, doi:10.3390/app122412768_

Round 1
Reviewer 1 Report
Deformed LSC liner collapse was very precisely analysed and a lot of parameters were taking into account (collapse point angle and arc radius after detonation, jet and slug velocity, jet velocity difference per time, etc.).
This paper is prepared very well.
Author Response
Dear Reviewer,
I really appreciate your comments. Based on comments from other reviewers, I have made a revised paper. If you any other comments that you want to add, please let me know.
Best Regards,
Reviewer 2 Report
The paper is the very interested, and deserved to be published after authors
will make small corrections:
1) Please add x and t ticks to Figs 3 & 4.
2) Please add all measurements of the copper lined LSCs and its speed in time
of the meeting with target.
3) Please convert all measurements in SI.
Author Response
Dear Reviewer,
I really appreciate your in-depth comments regarding the manuscript and it helps a lot. In order to answer your questions, I have revised the manuscript and please see below for my response per each question.
1) Please add x and t ticks to Figs 3 & 4.
>> The figures 3 and 4 are not an x-t diagram, but this is a schematic diagram representing a moment during the liner collapse or jet penetration. If I don't see your point, please direct me in a right path.
2) Please add all measurements of the copper lined LSCs and its speed in time
of the meeting with target.
>> The theoretical calculation is based on the liner collapse point where is invisible by the x-ray. In addition, the jet has to fly around several inches of distance inside the target before it got caught by the x-ray images. Which means that the calculated jet velocity won't match with the experimental jet velocity in the x-ray images since the jet velocity will decrease severely as it flies long distance. In addition, the jet speed of group A and random jet is not easy to observe due to the low image quality. Because of this limitation, the direct comparison between the two won't make any sense, and the calculation of the jet speed in the x-ray images didn't accomplished. Instead, we made assumptions that the jet tip and jet base in the x-ray images will be originated by the liner collapse start time (2 micro sec) and the liner collapse completion time (10 micro sec), and we compare the jet behaviors in this range in a newly added extra figure 9. This method still cannot give us a clear or direct comparison, but it gives us a good estimation of the penetration performance per each segment of jets. All these explanation has been added in the revised paper including the conclusion as well.
3) Please convert all measurements in SI.
>> I have added all the SI units in the revised paper.
I hope all the additions make sense to you.
Best Regards,
Reviewer 3 Report
Please, do not use style like ...In this paper, we empirically observe. Better is to use passive form like "empirically observations are presented..."
Please, provide more information about experimental setup, especially for construction of LSC (stand off, liner material, thickness, VOD of used explosive, dimension of LSC). All those parameters are relevant for efficiency and formed jet characteristic.
Also, target material properties are involved in interaction with jet penetration, so maybe it will be useful to provide some mechanical properties of UHMW.
Jet speed of the LSC is stated calculated. According to which model or formula are the calculation carried out and what are calculated values?
Please, could you provide some exact measured values of pictured jet in blocks, rather than relatively described relations between jets (much bigger or so).
In chapter "Theoretical analysis" Lim`s approach is described and results of that research. Could you connect yours described test and calculate parameters for present research and presented LSC in the paper. Please, statements in conclusion support with numbers.
Author Response
Dear Reviewer,
I really appreciate your in-depth comments regarding the manuscript and it helps a lot. In order to answer your questions, I have revised the manuscript and please see below for my response per each question.
Please, do not use style like ...In this paper, we empirically observe. Better is to use passive form like "empirically observations are presented..."
>> Yes. You are correct. This is an item that was changed by an English editor and I need to change it back to the original. In fact, I have found around 5 spots regarding this concern, and I have changed all of them. They are all included in the revised manuscript.
Please, provide more information about experimental setup, especially for construction of LSC (stand off, liner material, thickness, VOD of used explosive, dimension of LSC). All those parameters are relevant for efficiency and formed jet characteristic.
>> This is a good point. In order to answer this question, I added Table 1 in the revision. I have published the specs of 2000gr/ft LSCs (with the drawing and dimension included) before, and I added the reference in the table as well. The specs of LSCs is somewhat related with the manufacturer's right, I have tried not to step over someone's toe, and this is a bit of haze to me and was my concern. However, I have added much information about this, and hopefully this works.
Also, target material properties are involved in interaction with jet penetration, so maybe it will be useful to provide some mechanical properties of UHMW.
>> The newly added Table 1 will have the mechanical properties of UHMW. However, this is only for a static property (which is not closely related with this paper), but I hope this is good enough to see the general aspects of the target. I was not able to find any dynamic properties for this target material. Even if I found one, there is a high chance the dynamic property will not be as extreme as this paper.
Jet speed of the LSC is stated calculated. According to which model or formula are the calculation carried out and what are calculated values?
>> A set of equations are added in the paper in page 8. This equation is related with the Figure 4 and the results are in the Table 2.
Please, could you provide some exact measured values of pictured jet in blocks, rather than relatively described relations between jets (much bigger or so).
>> I revised the Figure 2 in a way to find the comparative length scale based on the max penetration of 5~6inches. I added a line scale right in the figure and this will help to understand the dimension of each figure.
In chapter "Theoretical analysis" Lim`s approach is described and results of that research. Could you connect yours described test and calculate parameters for present research and presented LSC in the paper. Please, statements in conclusion support with numbers.
>> The theoretical calculation is based on the liner collapse point where is invisible by the x-ray. In addition, the jet has to fly around several inches of distance inside the target before it got caught by the x-ray images. Which means that the calculated jet velocity won't match with the experimental jet velocity in the x-ray images since the jet velocity will decrease severely as it flies long distance. Because of this limitation, the direct comparison between the two won't make any sense. Instead, we made assumptions that the jet tip and jet base in the x-ray images will be originated by the liner collapse start time (2 micro sec) and the liner collapse completion time (10 micro sec), and we compare the jet behaviors in this range in a newly added extra figure 9. This method still cannot give us a clear or direct comparison, but it gives us a good estimation of the penetration performance per each segment of jets. All these explanation has been added in the revised paper including the conclusion as well.
I hope all the additions make sense to you.
Best Regards,
Round 2
Reviewer 3 Report
Thank you for accepting my observations and recomendation